# Medical students' knowledge of and attitudes towards LGBT people and their health care needs: Impact of a lecture on LGBT health

Raphaël Wahlen[1]*, Raphaël Bize[2‡], Jen Wang[1], Arnaud Merglen[3‡], Anne-Emmanuelle Ambresin[1,4]

1 Interdisciplinary Division for Adolescent Health (DISA), Lausanne University Hospital (CHUV), Lausanne, Switzerland, 2 Center for Primary Care and Public Health (Unisanté), University of Lausanne, Lausanne, Switzerland, 3 Division of General Pediatrics, Department of Child and Adolescent, University Hospitals of Geneva (HUG), Geneva, Switzerland, 4 Faculty of Biology and Medicine, University of Lausanne, Lausanne, Switzerland

☺ These authors contributed equally to this work.
‡ These authors also contributed equally to this work.
* raphael.wahlen@chuv.ch

**Data Availability Statement:** Data cannot be shared publicly because participants did not provide such consent at the time of participation. Furthermore, the study population is a small

## Abstract

### Objectives

Lesbian, gay, bisexual, and transgender (LGBT) adolescents have specific health care needs and are susceptible to health care disparities. Lack of skills and knowledge on the part of health care providers have a negative effect on their access to care and health outcomes. This study 1) explores the knowledge and attitudes of medical students regarding LGBT people, and 2) assesses the impact of a one-hour lecture targeting adolescent LGBT health needs.

### Methods

Fourth-year medical students attended a compulsory one-hour lecture on sexual orientation and gender identity development in adolescence, highlighting health issues. We created a questionnaire with items to elicit students' knowledge and attitudes about LGBT health issues. Students were invited to complete this questionnaire online anonymously one week before the lecture and one month after the lecture.

### Results

Out of a total of 157 students, 107 (68.2%) responded to the pre-intervention questionnaire and 96 (61.1%) to the post-intervention questionnaire. A significant proportion—13.7% of all respondents—identified as LGBT or questioning. Our results show that most medical students already show favorable attitudes towards LGBT people and a certain degree of knowledge of LGBT health needs. They demonstrated a large and significant increase in knowledge of LGBT health issues one month after the lecture.

number of participants who all know each other, and their responses to socio-demographics with skewed distributions make them indirect identifiers which may risk identification of study participants. Data protection laws have become even stricter in Switzerland in recent years, and currently, data availability procedures must be spelled out for ethical approval. Data are available to researchers who meet the criteria for access to confidential data by contacting the corresponding author, the cantonal ethics committee (secretariat.cer@vd.ch), the departmental research antenna (contact form: https://www.chuv.ch/fr/dfme/dfme-home/recherche/antenne-de-recherche-clinique/), or the institutional research center (contact form: https://www.chuv.ch/fr/crc/crc-home/).

**Funding:** The author(s) received no specific funding for this work.

**Competing interests:** The authors have declared that no competing interests exist.

## Discussion

A single one-hour lecture on sexual orientation and LGBT health issues may increase knowledge among medical students. Medical students and professionals should receive such training to increase their knowledge about LGBT patients as it, together with favorable attitudes, has the potential to improve health outcomes among this vulnerable population.

## Introduction

Lesbian, gay, bisexual and transgender (LGBT) adolescents are vulnerable to poor health and social outcomes because of marginalization, stigma and normative pressure against sexual and gender minorities [1,2,3,4,5,6]. They face personal and inter-personal challenges associated with the coming-out process, whereby many milestones are experienced as difficult [7,8]. Unlike other minorities, they cannot necessarily count on support from their parents or family and are therefore particularly vulnerable [9].

LGBT adolescents are targets of verbal and physical violence [10,11]. Thus, they experience greater psychological distress with higher levels of depression, anxiety, body image and eating disorders than the general adolescent population [12,13,14]. In Switzerland, the risk of suicide attempts are 2 to 7 times greater among sexual minority adolescents than heterosexual adolescents [15,16,17,18,19]. Internationally, the risk of suicide attempts is up to 10 times greater among transgender adolescents [20]. This is a real issue for paediatricians as more than half of those who attempt suicide do so for the first time before age 20 years [15,16,17]. LGBT adolescents are also 5 times more prone to substance use and to risky sexual behaviors [21,22,23,24].

The Society for Adolescent Health and Medicine encourages providers to incorporate the impact of these developmental processes (and understand the impact of potentially concurrent discrimination) when caring for LGBT adolescents [25]. Both the World Health Organization (WHO) and US Healthy People 2020 identified the poor health of LGBT persons as an area for improvement [26, 27]. Optimal provision of health care and prevention services to sexual and gender minorities requires providers to be sensitive to and informed about stigmatization, continued barriers to care access and the specific risk factors and health conditions in these populations [28,29]. Furthermore, each subgroup has specific health care needs which have to be known by health care providers to ensure quality of care [11,12].

Most often, health care providers are neither trained in nor sensitized to the health needs of LGBT people [11,12]. In addition, professionals generally find it difficult to discuss sexuality, all the more so if it is about sexual orientation or gender identity [30,31]. In this way, LGBT patients experience barriers in access to adequate healthcare due to a lack of specific knowledge and/or heterosexist attitudes on the part of health professionals. For example, heterosexist attitudes could lead to erroneous risk assessment of sexually transmitted infections and pregnancy as well as insufficient or improper use of screening tools. Such barriers can have a negative impact on the management, the treatment and finally the health of these patients [3,32].

One important strategy for improving knowledge and attitudes about LGBT people among providers is to train medical students during their medical studies in order to enable them to feel more comfortable when caring for these patients and to provide more adequate care. In the current literature, few studies have examined the knowledge and attitudes of medical students towards LGBT people, and even less have tested the impact of interventions among care providers. These studies—mostly from North America and Western Europe—show that medical students lack knowledge of LGBT healthcare needs, that they do not feel fully prepared to

care for these patients and that they are less comfortable with taking a sexual history and discussing sexual practices, with inadequate training reported as the main barrier to taking an adequate sexual history [33,34,35,36,37]. However, students in recent studies demonstrate mostly favorable attitudes toward LGBT people [37,38,39,40,41,42,43]. Still, data from other parts of the world show less favorable attitudes among medical students [44,45,46]. There is no such data for Switzerland. Finally, a few studies in North America and Europe have demonstrated the effectiveness of a short focused intervention in changing the knowledge, attitudes and comfort of medical students toward LGBT persons [40,41,47,48,49].

The aims of this study were 1) to assess the knowledge and attitudes of medical students in French-speaking Switzerland regarding LGBT people, and 2) to evaluate the impact of a compulsory one-hour lecture on adolescent LGBT health needs on these outcomes.

## Materials and methods

A compulsory one-hour lecture on sexual orientation and gender identity development during adolescence was offered to all fourth-year medical students at the Faculty of Medicine at the University of Lausanne in the fall semester 2016. The lecture focused on facts about health issues of LGBT adolescents and was given by a pediatrician experienced in adolescent health (AEA).

One week before the lecture, we explained the aims and the methods of our study directly to the students in the lecture hall. All students who completed both pre- and post-intervention questionnaires were eligible to take part in a draw for CHF 100 gift certificate from a large department store. Then an e-mail with a link to the online questionnaire was sent to all 4th-year students inviting them to complete the pre-intervention questionnaire before the course. Questionnaires were completed anonymously, but students were asked to provide the last five digits of their cell phone number, which allowed matching of pre- and post-intervention questionnaires. One month following the course, an e-mail with a link to the online questionnaire was sent to all students, asking students who completed the pre-intervention questionnaire to complete an identical post-intervention questionnaire.

The core of the questionnaire consists of 28 statements, with responses on a Likert scale ranging from 1 (*strongly agree*) to 5 (*strongly disagree*). These items were taken from several different scales like the Attitudes Towards Homosexuals Questionnaire (ATHQ) [49], Sex Education and Knowledge about Homosexuality Questionnaire (SEKHQ) [50], LGBT assessment scale [51], Genderism and Transphobia scale [52], and from studies with specific medical knowledge questions designed for medical students [46,47]. These items were selected to characterize student knowledge, attitudes and experiences along with a range of LGBT health issues. While the lecture focuses on health issues of LGBT adolescents, the statements from various scales did not focus specifically on adolescents, but the LGBT population generally. The items were translated from English into French by native French-speaking physicians for this study as no prior translations of these scales could be found.

The questionnaire also elicited demographic information from each student, including age, sex at birth, religion, sexual orientation, gender identity, as well as prior attendance at two other courses addressing LGBT health issues at the Faculty of Medicine.

The study protocol was submitted to the cantonal ethics committee which determined that the current study does not require evaluation according to federal and cantonal laws. Student participation was voluntary, and informed consent was obtained from all participants.

Statistical analyses were carried out using IBM SPSS Statistics 23 for PC. Exploratory factor analyses (EFA) were carried out for all 28 items. In order to increase the subject to item ratio, we used responses from all 203 pre- and post-intervention questionnaires in the EFA which

correspond to the standard factor loading cut-off of 0.40. Both the high Kaiser-Meyer-Olkin measure of sampling adequacy (.85) and the highly significant Bartlett's test of sphericity ($\chi^2(253) = 19334.49$, p < .001) indicate that factor analysis is appropriate for all 28 items. Factor extraction was carried out using the Principal Axis Factor method to better accommodate violations of normality. Although 8 factors had eigenvalues greater than 1, the point of inflection on the scree plot was between 3–5 factors. Since the factors were neither assumed nor shown to be uncorrelated (-.13 - -.46), an oblique rotation was implemented, yielding superior factor loadings.

A four-factor model was selected based on model fit indices and the interpretability of factors. Factor 1 (labeled Attitudes) includes 15 items, 3 of which with factor loadings below 0.40. Factor 2 (labeled Knowledge) included 5 items, 2 of which with factor loadings below 0.40. These five items with factor loadings below 0.40 were dropped in reliability testing for each factor: Factors 1–2 demonstrated good internal consistency—Attitudes (Cronbach's alpha = 0.89) and Knowledge (Cronbach's alpha = 0.82)—whereas Factors 3 (labeled Judgment with 4 items) and Factor 4 (labeled Experience with 4 items) demonstrated borderline internal consistency—Judgment (Cronbach's alpha = 0.68) and Experience (Cronbach's alpha 0.62). The items in each factor were summed and standardized on a scale from 0–100 with higher scores being more favorable to LGBT people.

These four standardized scores constituted the main dependent variables in subsequent analyses. ANOVA was used to check for differences in mean construct scores by dichotomous background variables for all respondents to the pre-intervention questionnaire (n = 107). Given skewed distributions for all construct scores, the non-parametric Wilcoxon signed rank sum test was used to assess case-level changes in construct scores for all those who reported having attended the fall lecture and completed both pre- and post-intervention questionnaires (n = 64). Cohen's d was calculated as a measure of effect size, with 0.20 considered small, 0.50 medium, and 0.80 a large effect size. As a checking exercise, change in construct scores of those who reported not having attended the fall lecture but still completed both pre- and post-intervention questionnaires (n = 22)—though not constituting a proper "control" group per se—were compared to the aforementioned "intervention" group (n = 64) using repeated measures general linear models.

## Results

Out of a total of 157 fourth-year students enrolled at the Faculty of Medicine, 107 (68.2%) responded to the pre-intervention questionnaire and 96 (61.1%) to the post-intervention questionnaire. Eighty-six (54.7%) students responded to both questionnaires and 31 (19.7%) to only one of the two questionnaires, for a total of 117 (74.5%) distinct respondents. Among the 86 students who completed both questionnaires, 64 (74.4%) reported having attended the lecture on adolescent LGBT health (Fig 1).

The socio-demographic background of the respondents is shown in Table 1. The mean age of the respondents—23 years—was comparable to that of the entire 4th-year class. Men constituted 32.5% and women 67.5% of the respondents, which means that only 61.3% of all the men in the lecture responded to the questionnaire compared to 83.1% of the women (p = .005). Twelve (11.7%) foreign students completed the questionnaire, with all foreign students in the lecture (14.6%) coming from other European countries via the Erasmus exchange program. Information on religion, sexual orientation and gender identity of respondents was obtained for the study group, but no such data are collected by the Faculty of Medicine for its student body. Among respondents, 16 (13.7%) students did not define themselves as heterosexual. No one defined themselves as transgender or other gender identity.

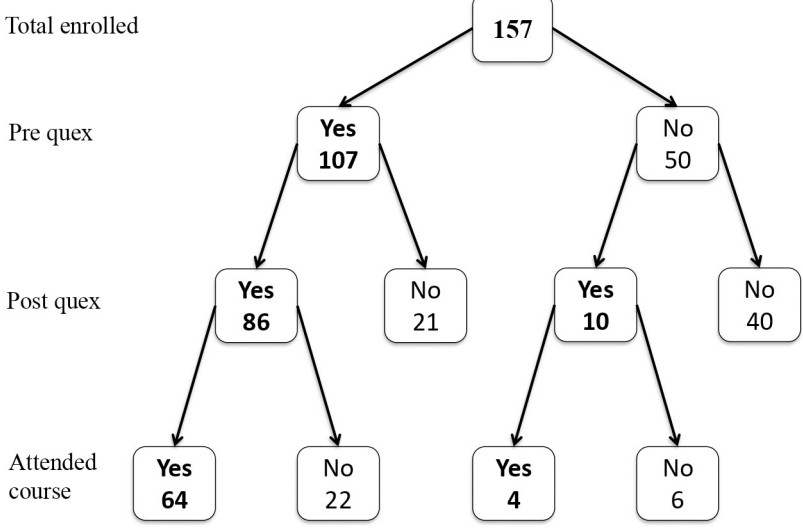

**Fig 1. Study participation flow diagram.**

Prior to fall 2016, fourth-year medical students had already had the possibility to participate in a class covering LGBT health. Indeed, there was a one-hour compulsory lecture on vulnerable populations—including migrants and LGBT people—the previous year, and some students also chose to participate in an elective mini-course (3 meetings of 45 minutes each) focusing on LGBT health and care. So, out of a total of 117 distinct respondents, 90 (76.9%) reported having attended a prior class covering LGBT health—85 (72.6%) the compulsory lecture on vulnerable populations and 21 (17.9%) the elective mini-course on LGBT health. Sixty-eight students reported having attended the fall 2016 lecture (70.8% of the 96 post-intervention respondents).

Table 2 shows all 28 items as organized into 4 factors after exploratory factor analysis as described in the Materials and Methods section.

Table 3 shows **pre-intervention** construct scores by background among respondents (pre-intervention, 107 respondents), on a scale from 0–100, with higher scores being more favorable to LGBT people. Overall, we observe high scores representing favorable attitudes toward LGBT people. We note no differences between male or female students in any of the four categories. We see a statistically significant difference in Knowledge scores between Swiss students and foreign students, with foreign students having a lower score (59.2 vs. 73.1, p = .04). Scores are similar for students by religion, but students with active religious practice have less favorable scores in both Attitudes (75.3 vs. 84.7, p = .05) and Knowledge (63.1 vs. 73.2, p = .08). We found similar scores for all categories between heterosexual and non-heterosexual students with a trend for greater Experience among non-heterosexuals (82.9 vs. 74.0, p = .10). Students who had taken any prior classes on LGBT health did not demonstrate statistically significantly higher scores on any of the four constructs.

Table 4 shows the evolution between pre- and post-lecture scores for each construct (Attitudes, Knowledge, Judgment and Experience) among the 64 respondents who attended the fall lecture and responded to both questionnaires. There were significant improvements in all four constructs; however, Knowledge is the only one which undergoes a large change with a Cohen's d of 0.84. Indeed, the average Knowledge score rose from 73.7 to 87.9 points on a standardized 0–100 scale, whereby 71.9% respondents demonstrated a higher score and 6.3% a

**Table 1. Background of the medical student respondents in Lausanne, Switzerland (n = 117).**

| Characteristic | Respondents (n = 117) n (%) | All 4th-year students (N = 157) n (%) |
|---|---|---|
| age (mean, SD) | 23.0 (1.87) | 23.6 (2.94) |
| sex at birth | | |
| male | 38 (32.5) | 62 (39.5) |
| female | 79 (67.5) | 95 (60.5) |
| nationality | | |
| Swiss | 91 (88.3) | 134 (85.4) |
| foreign | 12 (11.7) | 23 (14.6) |
| religion | | NA |
| Catholic | 39 (33.3) | |
| Protestant | 17 (14.5) | |
| other Christian | 4 (3.4) | |
| Jewish | 1 (0.9) | |
| Muslim | 4 (3.4) | |
| other religion | 6 (5.1) | |
| none | 36 (30.8) | |
| unknown/missing | 10 (8.6) | |
| religious practice | 15 (12.8) | NA |
| sexual orientation | | NA |
| heterosexual | 101 (86.3) | |
| bisexual | 6 (5.1) | |
| gay/lesbian | 5 (4.3) | |
| questioning | 3 (2.6) | |
| other | 1 (0.9) | |
| does not wish to respond | 1 (0.9) | |
| gender identity | | NA |
| male | 38 (32.5) | |
| female | 78 (66.7) | |
| does not wish to respond | 1 (0.9) | |

NA = not available

lower score post-lecture. On average across all constructs, 64.1% of respondents moved towards a higher score and 18.0% towards a lower score. However, comparable significant increases over time were observed also among those respondents who reported not having attended the course (data not shown).

## Discussion

### Favorable attitudes and knowledge (at baseline)

Similar to other recent studies in Western Europe, we found that a majority of medical student respondents show mostly non-rejecting attitudes towards LGBT people and demonstrate some knowledge about this population. However, it is unclear whether this is the result of prior courses at the Faculty of Medicine and/or due to social evolution. Favorable attitudes and knowledge towards LGBT health needs can be considered a positive development as medical settings often are known as being socially conservative [53].

**Table 2. Exploratory factor analysis of items measuring knowledge of, attitudes towards, and experiences with LGBT people pre- and post-class among medical students in Lausanne, Switzerland (n = 117).**

| Cat | Item | Cat factor | Factor 1 | Factor 2 | Factor 3 | Factor 4 |
|---|---|---|---|---|---|---|
| P | 26. School sex education programs should address all sexual orientations. | P | **-0,810** | -0,009 | -0,014 | -0,055 |
| P | 16. Changing an individual's sex (hormones and / or surgery) is against my moral values. | P | **0,801** | 0,051 | -0,131 | -0,001 |
| P | 15. Homosexual couples should be allowed to marry. | P | **-0,763** | -0,087 | -0,069 | 0,005 |
| P | 7. I think homosexuality is immoral. | P | **0,697** | -0,067 | -0,008 | -0,143 |
| A | 14. If I could choose, I would prefer not to provide care to a transgender person. | P | **0,647** | -0,105 | 0,119 | 0,110 |
| A | 10. If I could choose, I would prefer not to provide care to a gay, lesbian or bisexual person. | P | **0,623** | -0,139 | 0,062 | -0,085 |
| P | 17. Identifying as transgender should be considered a psychiatric illness. | P | **0,594** | 0,043 | 0,147 | -0,029 |
| P | 8. Groups that defend the rights of LGBT people are necessary. | P | **-0,580** | 0,100 | -0,137 | 0,015 |
| K | 13. Lesbian patients don't need a cervical smear as often as heterosexual women. | P | **0,509** | -0,067 | 0,093 | 0,135 |
| A | 9. As a physician, I think it is important to include questions about the personal and sexual life, sexual orientation and gender identity of my patients. | P | **-0,490** | 0,142 | -0,061 | -0,041 |
| P | 5. Gay couples should be allowed to adopt children | P | **-0,483** | -0,187 | 0,050 | 0,334 |
| P | 19. Imagining people of the same sex in intimate situations makes me uncomfortable.. | P | **0,453** | 0,059 | -0,049 | -0,131 |
| A | 20. I would feel uncomfortable examining and providing care to someone of my sex who is homosexual. | (P) | 0,367 | -0,001 | 0,063 | -0,177 |
| A | 24. I would be comfortable if my colleagues learned that I provide care to LGBT patients. | (P) | -0,275 | 0,116 | -0,053 | 0,169 |
| E | 11. When I meet a colleague or patient, I usually assume that he / she is heterosexual | (P) | 0,200 | 0,003 | -0,009 | -0,117 |
| K | 28. Gay and lesbian people have a higher prevalence of anxiety and depression compared to heterosexual people. | K | -0,045 | **0,876** | 0,125 | 0,203 |
| K | 27. LGBT adolescents are more likely to use alcohol, tobacco or other psychoactive substances than other adolescents. | K | -0,060 | **0,773** | 0,100 | 0,053 |
| K | 23. LGBT adolescents attempt suicide in the same proportions to those observed among heterosexual adolescents | K | 0,219 | **-0,582** | 0,107 | -0,079 |
| K | 1. LGBT adolescents have the same health needs as non-LGBT adolescents. | (K) | -0,141 | -0,252 | 0,161 | -0,051 |
| K | 25. Breast cancer can still occur after bilateral breast reduction surgery for transgender men. | (K) | -0,031 | 0,208 | -0,005 | -0,054 |
| P | 21. Homosexual men are generally effeminate and homosexual women generally masculine. | J | 0,039 | 0,158 | **0,696** | -0,237 |
| P | 3. Homosexual people can be identified by their appearance and their mannerisms. | J | 0,136 | 0,115 | **0,619** | -0,085 |
| K | 18. In our society, LGBT youth are currently well accepted, and therefore, there are no barriers in their access to medical care. | J | 0,085 | -0,170 | **0,515** | 0,064 |
| K | 6. Sex reassignment surgery is readily available for trans people and covered by health insurance. | J | 0,018 | -0,057 | **0,446** | 0,104 |
| E | 12. My interactions with LGBT people have positively influenced my perceptions towards LGBT people. | E | -0,073 | -0,042 | -0,187 | **0,562** |
| E | **4.** I socialize regularly with LGBT people in my everyday life.. | E | -0,025 | -0,022 | 0,032 | **0,500** |
| K | 2. The difference between sexual orientation and gender identity is clear to me. | E | -0,065 | 0,220 | -0,084 | **0,458** |
| P | 22. Being homosexual is a choice. | E | 0,086 | -0,163 | -0,034 | **-0,447** |
| Eigenvalues | | | 7,657 | 2,333 | 1,728 | 1,505 |
| % of total explained variance | | | 25,5 | 6,7 | 4,2 | 3,1 |

However, some subgroups demonstrate poorer knowledge and more negative attitudes. The lowest level of knowledge is found among foreign students (from other European countries via Erasmus) and among students describing themselves as religious observers. These students may come from or live in a more conservative environment with less favorable attitudes towards LGBT people. Contrary to the literature [38,39,45,46,54], we found no differences between male and female respondents along the four categories. However, as men were significantly less likely than women to respond to the questionnaire in the first place, these findings may be the result of self-selection, whereby male non-responders may have a different, potentially less favorable, profile on this issue.

**Table 3. Pre-intervention LGBT construct scores˚ by background among medical student respondents in Lausanne, Switzerland (n = 107).**

| Characteristic | Attitudes˚ mean (SD) | p | Knowledge˚ mean (SD) | p | Judgement˚ mean (SD) | p | Experience˚ mean (SD) | p |
|---|---|---|---|---|---|---|---|---|
| age | 83.4 (16.6) | 0,545 | 71.9 (20.4) | 0,780 | 68.9 (17.8) | 0,942 | 72.8 (19.3) | 0,059 |
| sex at birth | | 0,631 | | 0,413 | | 0,174 | | 0,375 |
| male | 82.9 (19.0) | | 74.3 (20.3) | | 65.4 (21.4) | | 72.8 (19.3) | |
| female | 84.0 (15.5) | | 70.8 (20.5) | | 70.5 (15.7) | | 76.4 (19.4) | |
| nationality | | 0,365 | | 0,044 | | 0,169 | | 0,562 |
| Swiss | 84.9 (15.0) | | 73.1 (20.7) | | 70.7 (15.5) | | 76.3 (18.8) | |
| foreign | 79.8 (28.2) | | 59.2 (18.6) | | 63.1 (23.3) | | 72.5 (23.4) | |
| religion | | 0,959 | | 0,844 | | 0,648 | | 0,596 |
| any religion | 83.4 (14.3) | | 72.0 (19.1) | | 69.2 (16.4) | | 74.3 (18.8) | |
| none | 83.6 (20.8) | | 72.9 (22.9) | | 67.5 (20.7) | | 76.4 (20.8) | |
| religious practice | | 0,049 | | 0,084 | | 0,821 | | 0,380 |
| yes | 75.3 (16.5) | | 63.1 (25.9) | | 67.9 (19.7) | | 71.0 (15.4) | |
| no | 84.7 (16.4) | | 73.2 (19.3) | | 69.0 (17.6) | | 75.9 (19.8) | |
| sexual orientation | | 0,936 | | 0,342 | | 0,646 | | 0,097 |
| heterosexual | 83.4 (16.5) | | 72.6 (20.5) | | 68.5 (18.6) | | 74.0 (19.4) | |
| other | 83.8 (18.2) | | 67.2 (19.5) | | 70.8 (11.7) | | 82.9 (17.3) | |
| any prior class on LGBT | | 0,816 | | 0,478 | | 0,318 | | 0,187 |
| yes | 83.6 (16.9) | | 72.7 (20.8) | | 69.8 (16.7) | | 76.6 (18.9) | |
| no | 82.3 (16.4) | | 69.3 (19.4) | | 65.8 (20.8) | | 70.8 (20.5) | |

˚ on a scale from 0–100, with higher scores being more favorable to LGBT

The high proportion (13.7%) of LGB (lesbian, gay and bisexual) or questioning students in this study greatly surpasses the proportion of self-identified LGB people in the general population (5%) [55]. This phenomenon may be explained by several factors, including greater representation of LGB people in higher education [56,57] and possibly higher rates of participation in this survey. This potentially significant proportion of non-heterosexual students should encourage the Faculty of Medicine to think about sexual and gender diversity internally and to develop a proactive approach on this issue as a number of medical students are themselves directly concerned. However, the fact that there were no differences in knowledge and attitudes along sexual orientation, means that sexual minority students also stand to benefit from education and training on sexual minority health needs.

Before the course, 76.9% of the respondents reported having attended a prior (compulsory and/or elective) class covering this issue. After this lecture, most of the respondents reported

**Table 4. Pre- vs. post-intervention scores˚ for knowledge of, attitudes towards, and experiences with LGBT people among medical students in Lausanne, Switzerland (n = 64).**

| Factor | Pre-class˚ mean (SD) | Post-class˚ mean (SD) | Z | p | d |
|---|---|---|---|---|---|
| Attitudes | 84.8 (13.6) | 86.8 (15.4) | -4.183 | < .001 | 0.14 |
| Knowledge | 73.7 (18.1) | 87.9 (15.7) | -5.289 | < .001 | 0.84 |
| Judgement | 69.8 (16.5) | 74.4 (18.8) | -2.479 | 0.01 | 0.09 |
| Experience | 77.0 (16.5) | 82.6 (16.8) | -3.135 | 0.002 | 0.34 |

˚ on a scale from 0–100, with higher scores being more favorable to LGBT people.

having attended a lecture on LGBT health by this point in their medical studies. However, it is important to keep in mind that possible selection bias among non-respondents and non-attendees may mean that the actual figure is lower. Attendance for both this and the previous compulsory lecture was around 70% whereas attendance for the elective was 18%. Compulsory lectures reach a much higher proportion of students than optional lectures. Therefore, core content should be transmitted via compulsory lectures.

Still, the findings show no differences between attendees and non-attendees of previous classes along the four categories at baseline. These results need to be interpreted cautiously as students may not have perfect recall of which lectures they took the previous year in a very dense medical teaching curriculum. It is also common practice for non-attending students to collect the course material afterwards to read on their own. This may be an explanation for changes over time observed among non-attendees.

## Improved knowledge (after lecture)

As seen in some previous studies [47,48], our results show that a one-hour lecture may improve knowledge and attitudes towards LGBT people among medical students. Indeed, the compulsory one-hour lecture on sexual orientation and gender identity development in adolescence—including specific health issues for LGBT adolescents—yielded improved scores among a majority of attendees in all categories one month after the intervention. In fact, improvements were seen post-lecture despite already favorable scores at baseline. It is not entirely surprising that the Knowledge category was the only one with a large effect size. Indeed, the course was focused on facts about the health issues of this population and not on changing perceptions or prejudices.

## Limitations

In addition to the limitations mentioned above, the following issues need to be taken into consideration when interpreting the findings. First, it is important to consider selection bias in both lecture attendance as well as survey participation. No data on attendance was collected on the actual day of the lecture, and background data from the Faculty of Medicine only allowed partial assessment of selection bias. Second, even though the response rate for each wave was satisfying, less than half the entire 4th-year class completed both questionnaires and attended the lecture. Although only students who completed the first questionnaire were asked to complete the second, 10 students who had not completed the first questionnaire completed the second anyway.

Third, due to its open-ended nature, the question on socio-economic status did not yield analyzable results. Fourth, two of the four categories–Judgment and Experiences–had borderline validity and consisted of disparate items. Better items need to be tested in future studies. Indeed, the items used may not capture more subtle and deep-seated aspects of gender (masculinity-femininity) and stigma which affect LGBT people. Lastly, the improvement in scores was seen in both those who had attended the lecture and those who had not. As the post-lecture survey took place several weeks after the lecture, many potential sources of contamination were possible during this long period. Students had access to the Powerpoint presentation online. The questionnaire itself, as well as some degree of media coverage on this topic at the time of the study, could have given rise to information and reflection.

## Implications

Training medical students on LGBT issues within the medical school curriculum is one possible way to increase student knowledge of their health needs, thereby contributing to the

ultimate goal of improving the health of LGBT adolescents. Future interventions need to evaluate how to best coordinate and train students on LGBT health care issues throughout their studies in order to best target and have a favorable impact on the different dimensions—i.e., knowledge, attitude, judgment and experience towards LGBT people. Using clinical vignettes, workshops or testimonials that elicit students' emotional response may be more effective in facilitating change in their attitudes and judgments as well as knowledge [Costa et al, 2007, Simmons 2012, Bales 1996].

Since there is growing evidence that a good patient-clinician alliance characterized by an empathic and positive therapeutic relationship improves quality of care and health outcomes for sexual minorities [32,58,59,60], training students and practicing doctors in communication skills and building therapeutic relationships between the patient and the clinician may constitute a key strategy.

The purpose of such endeavors is to enable future physicians to be sensitive to and informed about stigmatization, continued barriers to care and the specific risk factors and health conditions facing LGBT adolescents. Future research needs to assess the knowledge and attitudes towards LGBT adolescents among practicing pediatricians and general practitioners in Switzerland. This will help us understand whether being clinically exposed to LGBT adolescents increases positive attitudes of health professionals or whether LGBT adolescents receive poorer quality of care because of their sexual orientation. Such findings could point out needs which may be addressed in continuing education for primary care physicians.

## Conclusions

Using the medical teaching curriculum to sensitize future healthcare professionals to the needs of LGBT adolescents has the potential to improve health outcomes among this vulnerable population. Indeed, our study suggests that even a one-hour lecture can improve students' knowledge about LGBT health needs. However, knowledge is only one part of the equation in removing barriers and providing better care for a stigmatized group, and more work needs to be done to identify effective interventions that improve providers' attitudes towards this population.

## Supporting information

**S1 Appendix. Outline of one-hour lecture on sexual orientation and gender identity development.**
(DOCX)

## Author Contributions

**Conceptualization:** Raphaël Wahlen, Raphaël Bize, Anne-Emmanuelle Ambresin.

**Formal analysis:** Jen Wang.

**Funding acquisition:** Anne-Emmanuelle Ambresin.

**Investigation:** Raphaël Wahlen.

**Methodology:** Raphaël Bize, Jen Wang.

**Project administration:** Raphaël Wahlen.

**Supervision:** Jen Wang, Anne-Emmanuelle Ambresin.

**Writing – original draft:** Raphaël Wahlen.

**Writing – review & editing:** Raphaël Wahlen, Raphaël Bize, Jen Wang, Arnaud Merglen, Anne-Emmanuelle Ambresin.

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
