## [Decision Letter · Decision Letter 0]

6 Mar 2020

PONE-D-20-02563

Medical students’ knowledge of and attitudes towards LGBT and their health care needs: impact of a LGBT health lecture

PLOS ONE

Dear Dr. Wang,

Thank you for submitting your manuscript to PLOS ONE. After careful consideration, we feel that it has merit but does not fully meet PLOS ONE’s publication criteria as it currently stands. Therefore, we invite you to submit a revised version of the manuscript that addresses the points raised during the review process.

We would appreciate receiving your revised manuscript by Apr 20 2020 11:59PM. To enhance the reproducibility of your results, we recommend that if applicable you deposit your laboratory protocols in protocols.io, where a protocol can be assigned its own identifier (DOI) such that it can be cited independently in the future. For instructions see: http://journals.plos.org/plosone/s/submission-guidelines#loc-laboratory-protocols

We look forward to receiving your revised manuscript.

Kind regards,

Virginia E. M. Zweigenthal

Academic Editor

PLOS ONE

Additional Editor Comments (if provided):

Dear Authors,

In our assessment, this article is publishable subject to you making revissions as suggested by the two reviewers.

In addition i found two errors: line 143 'lecturer' should surely be 'students' and Line 146, 'anyways' is not correct English. I suggest 'anyway' or an equivalent.

We look forward to receiving your revisions.

Best wishes,

Dr Virginia Zweigenthal

Journal Requirements:

2) If materials, methods, and protocols are well established, authors may cite articles where those protocols are described in detail, but the submission should include sufficient information to be understood independent of these references (https://journals.plos.org/plosone/s/submission-guidelines#loc-materials-and-methods). In order to improve replicability and reproducibility, please provide additional details or supporting materials enabling other teachers and researchers to replicate your teaching intervention (in particular, about the lecture given in your study as an intervention). Please also provide, in addition to the version in the original language, a copy of your questionnaire given in Table 2 in English. If you include supporting materials, they should not be under a copyright more restrictive than CC-BY.

3)  We note that you have indicated that data from this study are available upon request. PLOS only allows data to be available upon request if there are legal or ethical restrictions on sharing data publicly. For information on unacceptable data access restrictions, please see http://journals.plos.org/plosone/s/data-availability#loc-unacceptable-data-access-restrictions.

4) We note that you have included the phrase “data not shown” in your manuscript. Unfortunately, this does not meet our data sharing requirements. PLOS does not permit references to inaccessible data. We require that authors provide all relevant data within the paper, Supporting Information files, or in an acceptable, public repository. Please add a citation to support this phrase or upload the data that corresponds with these findings to a stable repository (such as Figshare or Dryad) and provide and URLs, DOIs, or accession numbers that may be used to access these data. Or, if the data are not a core part of the research being presented in your study, we ask that you remove the phrase that refers to these data.

Reviewers' comments:

Reviewer's Responses to Questions

**Comments to the Author**

1. Is the manuscript technically sound, and do the data support the conclusions?

Reviewer #1: Yes

Reviewer #2: Yes

2. Has the statistical analysis been performed appropriately and rigorously? 

Reviewer #1: I Don't Know

Reviewer #2: Yes

3. Have the authors made all data underlying the findings in their manuscript fully available?

Reviewer #1: Yes

Reviewer #2: No

4. Is the manuscript presented in an intelligible fashion and written in standard English?

Reviewer #1: Yes

Reviewer #2: Yes

5. Review Comments to the Author

Reviewer #1: Dear author,

Your paper is interesting, and focuses on an important area in healthcare.

Small critiques: "Medicine" is misspelled in line 20.

Table two includes the questions asked of students in French. This must be changed to include either both French with the English translation, or just the English translation.

My expertise is medical education. From an educational point of view I think this is publishable- you seem to have touched on the important factors and limitations of this kind of intervention. It seems the majority of students had been exposed to some kind of teaching on LGBT beforehand, and it would be interesting to select out the students who had never had such an intervention as a subset of this cohort, to see if there were any interesting findings there- their data will give the most convincing effect of the intervention. I would recommend that you run the stats on this group as far as possible

The conclusion is over-reaching. This intervention, like most educational interventions, has the POTENTIAL to impact on health outcomes in this population. A one hour lecture (or any number of lectures, for that matter) does not guarantee implementation by the students in future, especially since the article admits that attitudes and feelings around the LGBT population were not explored in depth, but rather, knowledge about matters affecting their health outcomes was addressed. A student may thus have all the knowledge from the lecture but fundamentally opposed to the 'LGBT lifestyle'. So I would suggest a more conservative conclusion with a comment acknowledging that having the knowledge is only part of the story and that more work should be done on addressing preconceived ideas and judgments, because that issue can be even more of a barrier to care than a lack of knowledge.

Reviewer #2: As a PhD candidate and healthcare provider, I am glad to see more research emerging on the specific needs of sexual and gender minorities, especially in the healthcare settings.

The content and findings of this study are very relevant, for a variety of professionals and institutions. It is well-written and the literature findings outlines in the introduction are a good reflection of some of the major concerns of current sexual and gender minority health. The organization of the manuscript is clear and logical, as are the choice with regards to methodology and the conclusion based on the findings.

I suggest some minor revisions before publications.

1. Introduction

It might be interesting to very briefly elaborate what the negative impacts of 'heterosexist attitudes' are when it comes to treatment etc. (page 4, second paragraph) and why it is hence important to address them through interventions such as the one studied.

2. Terminology

The use of LGBT on its own is frowned upon within sexual and gender minority communities. Please consider using LGBT patients/people/adolescents etc. instead of just LGBT. This is done through most of the manuscript but not consistently. I.e., change 'Overall, we observe high scores representing favorable attitudes toward LGBT.' to 'Overall, we observe high scores representing favorable attitudes toward LGBT people.' (page 15, line 45, but found throughout the document, and especially in the tables).

In addition, please ensure to define LGB when first using this, as lesbian, gay, bisexual (LGB) (page 19, line 94).

3. Methodology

Kindly describe whether the instruments were translated into French (and how: translation alone or translation and back-translation? By whom? Were translations tested?) or whether previously-validated translations for the instrument were used.

4. References

The references are inconsistent – partly, APA is used, as are other styles. Please also ensure consistency in punctuation and capitalization.

5. Content

Were the foreign students asked where they came from? If so, this would be interesting to include, as references are made to changing, more favorable attitudes in Western Europe (page 19, line 75). As the students are European Erasmus students, seeing differences based on area of origin would be interesting.

6. Figures and tables

Figure 1 does not seem to be mentioned in the text.

Will the contents of table 2 be available in English, i.e. as an appendix?

I also suggest having a copy-editor review the manuscript to address minor grammatical and punctuation inconsistencies.

6. PLOS authors have the option to publish the peer review history of their article (what does this mean?). If published, this will include your full peer review and any attached files.

Reviewer #1: Yes: Chivaugn Gordon

Reviewer #2: Yes: Stephanie Haase

---

## [Author Response · Author response to Decision Letter 0]

2 Apr 2020

Reviewer #1: 

Small critiques: "Medicine" is misspelled in line 20. 

Thanks, we have corrected the spelling in the text.

Table two includes the questions asked of students in French. This must be changed to include either both French with the English translation, or just the English translation.

As noted above under Journal Requirements, we have provided the questions in English in Table 2 and moved the original questions in French to Appendix 2.

My expertise is medical education. From an educational point of view I think this is publishable- you seem to have touched on the important factors and limitations of this kind of intervention. It seems the majority of students had been exposed to some kind of teaching on LGBT beforehand, and it would be interesting to select out the students who had never had such an intervention as a subset of this cohort, to see if there were any interesting findings there- their data will give the most convincing effect of the intervention. I would recommend that you run the stats on this group as far as possible

You can see the findings comparing those students with prior exposure to LGBT health versus those without at baseline in the last row of Table 3. Contrary to our expectations, prior exposure did not appear to be associated with better scores both overall and for the compulsory general course on vulnerable groups and the specific elective on LGBT health individually. We summarize and discuss these findings in the Discussion section (1st complete paragraph on page 21).

The conclusion is over-reaching. This intervention, like most educational interventions, has the POTENTIAL to impact on health outcomes in this population. A one hour lecture (or any number of lectures, for that matter) does not guarantee implementation by the students in future, especially since the article admits that attitudes and feelings around the LGBT population were not explored in depth, but rather, knowledge about matters affecting their health outcomes was addressed. A student may thus have all the knowledge from the lecture but fundamentally opposed to the 'LGBT lifestyle'. So I would suggest a more conservative conclusion with a comment acknowledging that having the knowledge is only part of the story and that more work should be done on addressing preconceived ideas and judgments, because that issue can be even more of a barrier to care than a lack of knowledge.

Thank you for this note. We fully agree. We have adapted the conclusion (both in the abstract and the final conclusion) to reflect the reviewer’s point. We are aware of the multiple limitations of this study and the importance of interventions targeting preconceived ideas and judgments, as described at some length in the Implications sub-section of the Discussion (pgs 23-24).

Reviewer #2: 

As a PhD candidate and healthcare provider, I am glad to see more research emerging on the specific needs of sexual and gender minorities, especially in the healthcare settings.

The content and findings of this study are very relevant, for a variety of professionals and institutions. It is well-written and the literature findings outlines in the introduction are a good reflection of some of the major concerns of current sexual and gender minority health. The organization of the manuscript is clear and logical, as are the choice with regards to methodology and the conclusion based on the findings.

I suggest some minor revisions before publications.

1. Introduction

It might be interesting to very briefly elaborate what the negative impacts of 'heterosexist attitudes' are when it comes to treatment etc. (page 4, second paragraph) and why it is hence important to address them through interventions such as the one studied.

Thank you for this suggestion. We have mentioned faulty risk assessment and use of screening tools as examples in the text.

2. Terminology

The use of LGBT on its own is frowned upon within sexual and gender minority communities. Please consider using LGBT patients/people/adolescents etc. instead of just LGBT. This is done through most of the manuscript but not consistently. I.e., change 'Overall, we observe high scores representing favorable attitudes toward LGBT.' to 'Overall, we observe high scores representing favorable attitudes toward LGBT people.' (page 15, line 45, but found throughout the document, and especially in the tables).

In addition, please ensure to define LGB when first using this, as lesbian, gay, bisexual (LGB) (page 19, line 94).

Great remark, thank you very much. We have made the corrections in the text.

3. Methodology

Kindly describe whether the instruments were translated into French (and how: translation alone or translation and back-translation? By whom? Were translations tested?) or whether previously-validated translations for the instrument were used.

We address these points now in the text. The items were translated by native French-speaking physicians on our team (translation alone) as we could not locate any prior translations of these scales. 

4. References

The references are inconsistent – partly, APA is used, as are other styles. Please also ensure consistency in punctuation and capitalization.

We are using Vancouver style for the references. We have checked the references for consistency, but thankfully, many journals are now able to import the references automatically, thereby correcting small errors and inconsistencies in the process. 

5. Content

Were the foreign students asked where they came from? If so, this would be interesting to include, as references are made to changing, more favorable attitudes in Western Europe (page 19, line 75). As the students are European Erasmus students, seeing differences based on area of origin would be interesting.

No, the students were not asked to specify their country of origin in the questionnaire. 

6. Figures and tables

Figure 1 does not seem to be mentioned in the text. 

Thanks, we have added it in the text.

Will the contents of table 2 be available in English, i.e. as an appendix? 

As mentioned in our responses to Journal Requirements and Reviewer #1, we have used the English items in Table 2 and moved the French items actually used in the survey to Appendix 2.

I also suggest having a copy-editor review the manuscript to address minor grammatical and punctuation inconsistencies.

You’re absolutely right. This manuscript has undergone several revisions coordinated by the first author who is a native French speaker, and the co-author who is a native English speaker has now combed the manuscript in a final proofread.

---

## [Decision Letter · Decision Letter 1]

2 Jun 2020

Medical students’ knowledge of and attitudes towards LGBT and their health care needs: impact of a lecture on LGBT health

PONE-D-20-02563R1

Dear Dr. Wang,

We are pleased to inform you that your manuscript has been judged scientifically suitable for publication and will be formally accepted for publication once it complies with all outstanding technical requirements.

With kind regards,

Virginia E. M. Zweigenthal

Academic Editor

PLOS ONE

Additional Editor Comments (optional):

Reviewers' comments:

Reviewer's Responses to Questions

**Comments to the Author**

1. If the authors have adequately addressed your comments raised in a previous round of review and you feel that this manuscript is now acceptable for publication, you may indicate that here to bypass the “Comments to the Author” section, enter your conflict of interest statement in the “Confidential to Editor” section, and submit your "Accept" recommendation.

Reviewer #2: All comments have been addressed

2. Is the manuscript technically sound, and do the data support the conclusions?

Reviewer #2: Yes

3. Has the statistical analysis been performed appropriately and rigorously? 

Reviewer #2: Yes

4. Have the authors made all data underlying the findings in their manuscript fully available?

Reviewer #2: Yes

5. Is the manuscript presented in an intelligible fashion and written in standard English?

Reviewer #2: Yes

6. Review Comments to the Author

Reviewer #2: Thank you very much for taking my comments into consideration.

The only final minor revision is the terminology of LGBT people on page 1, should this ever be used for publishing purposes. Also, and I am not sure if this is possible at this point, adding the word 'people' or 'patients' to the title. (i.e. 'Medical students’ knowledge of and attitudes towards LGBT people and their health care needs: impact of a lecture on LGBT health'

7. PLOS authors have the option to publish the peer review history of their article (what does this mean?). If published, this will include your full peer review and any attached files.

Reviewer #2: Yes: Stephanie Haase

---

## [Editor Report · Acceptance letter]

5 Jun 2020

PONE-D-20-02563R1 

Medical students’ knowledge of and attitudes towards LGBT people and their health care needs: impact of a lecture on LGBT health 

Dear Dr. Wang:

I'm pleased to inform you that your manuscript has been deemed suitable for publication in PLOS ONE. Congratulations! Your manuscript is now with our production department. 

Kind regards, 

on behalf of

Dr. Virginia E. M. Zweigenthal 

Academic Editor

PLOS ONE